# Natural Root Cellular Variation in Responses to Osmotic Stress in *Arabidopsis thaliana* Accessions

**DOI:** 10.3390/genes10120983

**Published:** 2019-11-29

**Authors:** Wendy Cajero-Sanchez, Pamela Aceves-Garcia, María Fernández-Marcos, Crisanto Gutiérrez, Ulises Rosas, Berenice García-Ponce, Elena R. Álvarez-Buylla, Maria de la Paz Sánchez, Adriana Garay-Arroyo

**Affiliations:** 1Plant Molecular Genetic, Epigenetic, Development and Evolution Laboratory, Ecology Institute, National Autonomous University of Mexico, 3er Circuito Ext. Junto a J. Botánico, Ciudad Universitaria, UNAM, Mexico City 04510, Mexico; wendycs@ciencias.unam.mx (W.C.-S.); pamela.aceves@biologie.uni-freiburg.de (P.A.-G.); bgarcia@ecologia.unam.mx (B.G.-P.); elenabuylla@protonmail.com (E.R.Á.-B.); 2Biological Science Postgraduate, National Autonomous University of Mexico, Av. Universidad 3000, Coyoacán, Mexico City 04510, Mexico; 3Biochemical Science Postgraduate, National Autonomous University of Mexico, Av. Universidad 3000, Coyoacán, Mexico City 04510, Mexico; 4Molecular Biology Center Severo Ochoa (CSIC-UAM), Nicolás Cabrera 1, Cantoblanco, 28049 Madrid, Spain; fernanma@itacyl.es (M.F.-M.); cgutierrez@cbm.csic.es (C.G.); 5Botanical Garden, Biology Institute, National Autonomous University of Mexico, Mexico City 04510, Mexico; urosas@ib.unam.mx; 6Complexity Science Center (C3), National Autonomous University of Mexico, Mexico City 04510, Mexico

**Keywords:** osmotic stress, *Arabidopsis* accessions, root morphology, plasticity, natural variation

## Abstract

*Arabidopsis* naturally occurring populations have allowed for the identification of considerable genetic variation remodeled by adaptation to different environments and stress conditions. Water is a key resource that limits plant growth, and its availability is initially sensed by root tissues. The root’s ability to adjust its physiology and morphology under water deficit makes this organ a useful model to understand how plants respond to water stress. Here, we used hyperosmotic shock stress treatments in different *Arabidopsis* accessions to analyze the root cell morphological responses. We found that osmotic stress conditions reduced root growth and root apical meristem (RAM) size, promoting premature cell differentiation without affecting the stem cell niche morphology. This phenotype was accompanied by a cluster of small epidermal and cortex cells with radial expansion and root hairs at the transition to the elongation zone. We also found this radial expansion with root hairs when plants are grown under hypoosmotic conditions. Finally, root growth was less affected by osmotic stress in the Sg-2 accession followed by Ws, Cvi-0, and Col-0; however, after a strong osmotic stress, Sg-2 and Cvi-0 were the most resilience accessions. The sensitivity differences among these accessions were not explained by stress-related gene expression. This work provides new cellular insights on the *Arabidopsis* root phenotypic variability and plasticity to osmotic stress.

## 1. Introduction

Phenotypic plasticity is an important property that allows plants to respond to a wide range of different environments. This feature is influenced by internal and external conditions that modify developmental processes. *Arabidopsis thaliana* (hereafter *Arabidopsis*) is widely distributed around the northern hemisphere and consequently is subjected to diverse environmental conditions, generating different natural variants called accessions [1]. The *Arabidopsis* accessions are an important genetic resource to identify mechanisms underlying plant development and stress tolerance as plant genotypes are constantly shaped by biotic and abiotic factors [2]. These phenotypic and genetic variations have enabled the characterization of responses in *Arabidopsis* natural variants using a range of different approaches [3,4,5,6,7,8].

Water deficit is an abiotic stress that affects plant development and productivity. Water availability can be altered by changes in solute concentration (i.e., sugars, salt, inorganic cations and anions) during drought, cold stress, and freezing [9,10]. Some phenotypic and genetic analyses have identified tolerant accessions that can be useful to study the traits related to water deficit tolerance [11,12] (for a review see [13]).

Although stress affects the whole organism [14,15,16,17], leaves and roots display different responses in order to reduce water loss and promote water foraging for survival [18]. Accordingly, plant roots act on the frontlines by sensing the water deficit, adjusting osmotic homeostasis, forcing water entrance, and avoiding water loss through the accumulation of compatible solutes inside the tissues [19,20]. This rapid root response allows increased water uptake to maintain cellular turgor and reduce the negative effects in the leaves; however, when the stress becomes more severe, the growth in all tissues is highly compromised and it can cause the plant’s death [18,21,22]. All these cues are later communicated to the shoot, which might respond with reduced growth rates, stomata closure, or rapid senescence, but the root is the main organ that reads and responds to the water availability status [19,23]. Therefore, the root system not only represents a key model for studying water stress, but it is highly relevant to characterizing how the whole plant figures out strategies to face, tolerate, and recover from water stress. It is important to understand how these strategies have diverged in natural populations to deepen the genetic basis of the plant’s responses to drought stress.

The *Arabidopsis* root system is composed of primary and lateral roots with identical radial organization. In the primary root three distinct zones are distinguished by their abilities to proliferate, elongate, or differentiate [24,25,26]. The proliferative zone is at the root apical meristem (RAM), which contains the stem cell niche (SCN) that is formed by four different sets of stem cells (also called initial cells) that yield all root cell types [27]. These initial cells surround an organizer center called the quiescent center (QC) with very low mitotic activity and the capacity to produce short-range signals that are important for maintaining the initial cells in an undifferentiated state [28,29,30]. Additionally, the RAM can be subdivided in two domains: the proliferation domain (PD) and the transition domain (TD). In the former, cells proliferate for 4–6 cycles and maintain a relatively small size, whereas in the TD, cells have a lower proliferation rate and they start to enlarge [24,26]. The cells that stop proliferating and elongate anisotropically at very fast rates are confined to the elongation zone (EZ), whereas in the differentiation zone (DZ) cells acquire their final characteristics [25,26]. Proliferation and differentiation are two interlinked processes in which the cells that are produced in the meristematic region are then displaced from it to the elongation zone towards the differentiation zone. It has been shown that both processes contribute to the final organ size [31,32].

The imminent and drastic environmental changes caused by global warming and climate change have drawn global attention to understanding how plants cope with water deficit and osmotic stresses. Despite the vast literature dealing with the issue, little attention has been paid to the plant organ directly facing the stress on the frontline: the root. Therefore, in this study we decided to use nonionic solutes such as mannitol or sorbitol in order to evaluate root responses to water deficit, thereby changing the osmotic potential without adding ionic effects. We analyzed morphological alterations of root cells in response to hyperosmotic shock stress treatments in 15 *Arabidopsis* accessions, some of them characterized as salt-tolerant accessions based on their responses in aerial tissues [12]. We found that hyperosmotic stress inhibits root cell proliferation and elongation but does not interfere with QC identity or SCN morphology. Furthermore, under hyperosmotic stress, cortical and epidermal cells swelled and displayed a premature transition from the TD to the EZ and from the EZ to the DZ in all accessions. Interestingly, cell swelling occurred when the plant was subjected to a rapid osmotic shock treatment, with either an increase or decrease of solutes in the growth medium. The phenotypic primary root analysis revealed that root growth of the accessions Sg-2 and Ws was less affected by osmotic stress treatments than root growth of Cvi-0, which had an intermediate effect; whereas Col-0 and L*er* accessions were most severely affected. In addition, Sg-2 followed by Cvi-0 were shown to be the most resilient accessions in their recovery from strong hyperosmotic stress to control conditions. Unexpectedly, we did not find a correlation between the resiliency and the expression of different osmotic stress-related genes, suggesting that their increase in gene expression is not necessary to induce plant resilience.

## 2. Materials and Methods

The *Arabidopsis thaliana* accessions used in this work were: Büchen (Bch-4; ID: SJA26800), Buchschlag (Bu-5; ID:SJA02900), Burren (Bur-0; ID: SJA04400), Llagostera (Ll-1; ID: SJA33200), Schwieggershausen (Sh-0; ID: SJA21600), Sankt Georgen (Sg-2; ID: SJA21500), Wildbad (WI-0; ID: SJA25100), and Zurich (Zu-0; ID: SJA26400) from Riken Institute, Yokohama, Japan and Cape Verde Islands (Cvi-0; ID: N1096), Frankfurt (Fr-2; ID: N1168), HR (HR-5; N22205) and Tabor (Ta-0; ID: N1548) from Nottingham, England *Arabidopsis* Stock Centre (NASC). Columbia (Col-0), Landsberg *erecta* (L*er*) and Wassilewskija (Ws) were accessions routinely used in our laboratory for more than 15 years.

### 2.1. Plant Growth Conditions

Seeds from different *Arabidopsis* accessions were disinfected with 20% sodium hypochlorite and 0.01% of Tween 20 for 15 min and stratified at 4 °C for 5 days under dark conditions, and sown on square Petri dishes containing MS medium (0.2 × Murashige and Skoog salts (MP Biomedicals; Irvine, CA, USA), 0.05% MES (Sigma-Aldrich; St. Louis, MO, USA), 1% sucrose (Sigma-Aldrich), and 1% agar (Becton, Dickinson and Company; Franklin Lakes, NJ, USA)), at pH = 5.6. For osmotic treatments, five days after sowing (5 dps) seedlings were transferred to MS medium (Control) or MS supplemented with concentrations of mannitol or sorbitol as indicated in each case and grown for 24 h. For the recovery assays, 5-dps seedlings were grown in MS medium with 400 mM of mannitol (changing the plants to a new medium with 400 mM of mannitol every week). Afterwards, the seedlings were returned to control conditions (MS medium) for 10 days to finally transfer them to soil for 7 days. For the hypoosmotic assay, 5-dps seedlings were transferred for one day to hyperosmotic stress conditions (300 mM mannitol) and then returned to control conditions for another day. In all cases plants were grown in a chamber at 22 °C under long-day (LD; 16 h light/8 h dark) conditions with a light intensity of 110 µm^−2^ s^−1^.

### 2.2. Osmotic Potential Measurement

To measure osmotic potential (ψπ), we used the vapor-pressure osmometer (VPO) Wescor, model VAPRO Model 5600 (ELITech group; Puteaux, France). The instrument has a small depression where a filter paper disk is filled with 10 L of the solution to measure (100 mM, 200 mM, and 300 mM of sorbitol and 100 mM, 200 mM, 300 mM, and 400 mM of mannitol).

### 2.3. Pseudo-Schiff Assay

For root cellular analysis, the roots of 5-dps seedlings grown in MS medium were transferred for one day to MS medium (control) or to MS supplemented with 300 mM of mannitol, and then were fixed and stained according to a modified Truernit protocol [33]. This was done as follows: seedlings were fixed in a solution of 50% methanol and 10% acetic acid at room temperature (this can be done for 30 min up to two weeks). After fixation, roots were washed three times with distilled water and then incubated for 30 min in 1% periodic acid. After the periodic acid, plants were washed three times with distilled water and placed for 2 h in 0.18 M sodium bisulfite, 0.15 N hydrochloric acid, and 100 μg/mL propidium iodide at room temperature. Seedlings were washed again three times with distilled water and placed in Hoyer’s solution (80% chloral hydrate and 10% glycerol) for microscopy observation.

### 2.4. Microscopy Visualization

Root tissues were visualized using Nomarski optics under an Olympus BX60 microscope with a dry 40× objective and photographed with an Evolution MP COLOR camera of Media Cybernetics. Confocal images were acquired using the Olympus FV 1000 microscopy with an oil immersion 40× objective.

### 2.5. Kinematic Analysis

For all the quantitative cellular analysis, cell size and root domains and zones, were obtained using Fiji software [34]. The data were analyzed as previously described [35]. The growth rate and root length of each accession were obtained by marking the position of the root tip every 24 h on the back of the plate, the results of which were digitalized and measured using Fiji software. The cell size profile along the apical–basal axis of the root was obtained by measuring each cortex cell length along the cell file from the QC (cell 1) up to the fully mature zone (20 or more cells after the cortical cell nearest to the epidermal cell with the first hair root). The characterization of root domains and zones was done using a method based on double mobile linear regressions of cell length distributions along the root longitudinal axis, as described by multiple structural change algorithm (MSC) [36].

### 2.6. RNA Extraction and Quantitative RT-PCR

Five-days-old Col-0, Cvi-0, and Sg-2 seedlings were transferred to control or 300 mM mannitol supplemented media for 8 h. Total RNA was extracted from the whole roots using the RNeasy Plant Mini Kit (QIAGEN; Venlo, The Netherlands). Concentration and integrity of the extracted RNA were tested using a NanoDrop 2000c spectrophotometer (Thermo Scientific; Waltham, MA, USA) and bleach agarose gel electrophoresis (Aranda et al., 2012). RNA was then reverse-transcribed into cDNA with SuperScript III First-Strand Synthesis SuperMix (Invitrogen; Carlsbad, CA, USA). RT-qPCR was performed with SYBR Select Master Mix (Thermo Scientific) using the ΔΔ_Ct_ method. *UPL7* (AT3G53090), *PDF2* (AT3G22480), *RNAH* (AT4G00660) and AT5G15710 were used as reference genes. The primer sequences used in this study are listed in Appendix A. Each experiment was performed with three biological replicates.

### 2.7. Geometric Morphometric Analysis

The seedlings were grown for five days under control conditions and then transferred to either control, 100, 200, or 300 mM of mannitol and plates were scanned at 800 dpi resolution. For geometric morphometric analysis we used the Shape Model Toolbox software [37] implemented for roots as RootScape [38]. The model was made out of 20 landmarks as follows: one landmark at the base of the root, one at the tip of the root, two at the locations of the first and last elongated lateral roots along the primary root, and two at the widest points of elongated lateral roots at each side of the primary root. A total 14 pseudo-landmarks were placed evenly spaced between the landmarks; all the landmarks built a polygon that captured the convex hull-shape of the root architecture. This allometric model considered the length of the primary root, the branching pattern, and the angle of the primary root. Using the landmark data, a geometric morphometric principal component analysis was done with Procrustes for rotation and translation to the centroids in order to align the shapes, but without size normalization. Here we only showed the three principal components (PCs) that capture 93.7% of the variation as an arbitrary cutoff.

## 3. Results

### 3.1. Hyperosmotic Shock Stress Treatment Affects Primary Root Length, Proliferation, and Differentiation of Root Meristem Cells in the Arabidopsis Col-0 Accession

To study the effect of hyperosmotic stress conditions in *Arabidopsis* root growth we used the Col-0 accession to optimize a stress treatment transferring 5-dps seedlings to plates with or without different hyperosmotic stress conditions (see Appendix A for the experimental setup) and measured the length of the primary root. In order to avoid ionic stress, we used compatible solutes such as mannitol or sorbitol at concentrations of 100, 200, and 300 mM (Figure 1A and Appendix A). The root growth in different osmotic stress conditions using mannitol or sorbitol exhibited concentration-dependent inhibition, displaying a slightly stronger effect when using mannitol (Figure 1A and Appendix A). Previous studies in *Arabidopsis* seedlings have shown that an osmotic potential of −0.23 to −0.51 MPa is a moderate stress while a potential of −0.8 to −1.2 MPa represents a high stress level [39]. Therefore, we decided to use a high mannitol concentration (300 mM, −0.9361 MPa; Appendix A) to characterize how it affects root development. With this condition, the primary root length was inhibited after 24 h of treatment, reaching ~70% inhibition as compared to the control treatment (Appendix A and Figure 3A), which is consistent with previous reports [39,40,41].

Root length depends on the balance between cell proliferation and cell elongation, both processes related to RAM homeostasis. Generally, a high proliferation rate at the RAM produces more cells that are able to elongate and differentiate, resulting in a high growth rate [42]. To find out if the reduction in root length was due to defects in cell proliferation or in cell elongation and differentiation, we performed quantitative cellular analysis based on cell length measurements along the file of cortical cells in roots immediately after 24 h of osmotic stress in 300 mM mannitol. For this, we counted and measured cell length in the proliferative and the transition domains of the RAM and elongated cells and elongation size in the DZ and EZ, respectively. We observed that the cell number of the RAM (Figure 1C,D), as well as the length of the completely elongated cells (Figure 1E) and the EZ size (Figure 1C and Appendix A) were reduced under the osmotic stress condition in Col-0, explaining the root length reduction in this stress condition. Moreover, the RAM size reduction was due to a decrease in cell number in the PD (Appendix A) and a premature transit towards the elongation zone as the length of the cells in the TD in the osmotic stress condition was smaller than in control conditions; thus, they did not reach the suitable size before transiting to the elongation zone (Appendix A). Contrary to this, we could not find differences in the PD cell size or in the number of TD cells between control and stress treatments (Appendix A). The decrease in the PD number correlated with fewer proliferating cells and was observed in a gradual decrease of *CYCB1*; *1DB-GUS* (a proliferation marker of G2/M phase transition of the cell cycle [43]) from the first day of treatment, reaching low levels at 3 days (Figure 2B).

Under control conditions, cell length increased in the EZ and reached almost the final length in the maturation zone where the cells differentiate and acquire their final attributes (i.e., root hairs in epidermal cells) [25,44,45]. In contrast, in osmotic-stressed roots, the cells were smaller in the TD when they transited to the DZ and had shorter fully elongated cells in the EZ, altering the elongation zone size (Figure 1E, Appendix A). In consequence, roots developed hairs nearer to the QC (Figure 1C and Figure 2A, inset 1). Thus, root growth rate was drastically affected by osmotic stress conditions (Figure 1B) and primary root length reduction of Col-0 under osmotic stress conditions depended on altering both cell proliferation and differentiation.

### 3.2. Stem Cell Niche Organization Is Refractory to Hyperosmotic Stress Conditions

The RAM cell proliferation depends on the cells that are produced in the SCN, the cells that are proliferating, and the cells that transit to the EZ. Therefore, we addressed whether the osmotic stress also affects the SCN and QC identity and maintenance. Interestingly, we observed that, under hyperosmotic stress conditions, roots of the Col-0 accession showed normal SCN morphology and QC identity, revealed by expression of p*WOX5-GFP* and p*SCR-GFP* reporter markers (Appendix A) [46,47] and indicating that osmotic stress only affects cell proliferation in the RAM but not the SCN morphology or QC identity.

### 3.3. Hyperosmotic Stress Conditions Result in Swelling of the Epidermis and Cortex Cells of the Col-0 Accession

Under acute hyperosmotic treatments, root cells formed what we have called the stress zone (SZ), where the impact of the stress condition was undoubtedly observed (Figure 2A, insets 1 and 4). The SZ was observed in both 200 mM or 300 mM either of mannitol or sorbitol after 1 day of stress treatment in Col-0 (Figure 2A and Appendix A). Under osmotic stress the cells transited more rapidly to the elongation zone and prematurely acquired characteristics of differentiated cells in the epidermis (i.e., root hairs); although these premature cells were smaller than the cells with root hairs in control conditions (compared insets 2 and 4 in Figure 2A). We also observed a radial expansion mainly on epidermal and cortex cells in the SZ in different hyperosmotic stress conditions (Figure 2, inset 2 and Appendix A). It is noteworthy that when roots were kept under the stressful condition for one day, cells within the SZ continued growing as they transitioned to the EZ, but they did not reach the cell length of the DZ of control conditions (Figure 2A, compared insets 1 and 2). Finally, the cells that were fully elongated before the stress treatment had the same cell size as control cells (Figure 2A, compared insets 2 and 3).

### 3.4. Root Growth of Arabidopsis Natural Accessions Is Differentially Affected by Hyperosmotic Stress Conditions

With more than 7000 accessions, *Arabidopsis* displays a large range of phenotypic natural variation, possibly reflecting adaptations to the diversity of environments where they were collected. Previous reports have shown that root architecture and growth of several *Arabidopsis* accessions have wide variability in standard conditions [48]. In order to analyze in detail how the root architectural plasticity of *Arabidopsis* accessions is affected under osmotic stress growth conditions, we selected 15 accessions with contrasting root growth and analyzed their developmental patterns under 300 mM mannitol. As can be seen in Figure 3A, roots were dramatically shorter in all the accessions with 62–24% growth reduction under stress treatment as compared to control conditions. Yet, the response was variable among accessions. L*er*, Ta-0, Bch-4, Fr-5, and Col-0 were strongly affected with only 30–24% growth reduction (highly sensitive group), followed by Ll-1, Sh-0, Bu-5, Wl-1, and Cvi-0 with 38–32% growth reduction (moderately sensitive group), and by Bur-0, HR5, Ws, and Sg-2 that were the least affected with 62–41% growth reduction (mildly sensitive group) (Figure 3A). According to the root growth sensitivity of these accessions to osmotic stress, we selected five accessions (Col-0, Cvi-0, L*er*, Sg-2, and Ws) as representatives of each group to characterize in detail the effects of osmotic stress in root growth and performed a geometric morphometric (landmark-based principal component analysis) RootScape analysis [49]. For this analysis, the parameters selected were the primary root length, the angle of lateral root growth, and the locations of the first and the last lateral roots in the primary root as landmarks (Figure 3B). The PCA analysis showed that the primary root length explains almost 90% of the variation found among accessions under the four growth conditions (Figure 3C,D). This was surprising since, in high-salt stress, which also causes a negative osmotic potential, lateral root development is much more sensitive than primary root development [50]. Furthermore, dose-response curves to different mannitol concentrations (100, 200, and 300 mM) showed that Sg-2 was slightly affected by the 100 mM mannitol treatment and, interestingly, in two accessions (Cvi-0 and Sg-2), the root size did not change from 200 to 300 mM of mannitol, and in Ws root size did not change from 100 to 200 mM of mannitol (Figure 3E). These data indicate that root response and tolerance to increasing concentrations of osmotic stress are background dependent but does not depend on the initial size of the primary root.

All of these five accessions had shorter roots in the stress treatment compared to control conditions (Figure 4A,B); however, they continued to grow but at different rates; Col-0, Cvi-0, and L*er* were more severely affected than Sg-2 and Ws (Figure 4C and Appendix A). As occurs for Col-0 in the control condition, it could be expected that plants with longer roots would have a higher root growth rate, but this was not the case for Ws, which had a reduced root growth rate but medium root length (see the differences between L*er* and Ws). This could be the result of a smaller number of cells in the RAM but larger fully elongated cells, as can be observed when comparing the accessions Ws and L*er* (Figure 4D,E).

Additionally, quantitative cellular analysis showed that the decrease in root growth in all the accessions under osmotic stress conditions is explained by the combined developmental effects of a smaller meristem cell number and shorter cell length in the EZ as compared to control conditions, as shown in Col-0 (Figure 1D,E and Figure 4). In fact, the RAM cell number of all studied accessions was affected by the stress condition; Sg-2, Ws, and Cvi-0 being less affected than Col-0 and L*er* accessions (in that order) (Appendix A). In addition, the RAM cell number was the result of the sum of both PD and TD cell numbers and, similar to Col-0, the number of the PD cells explained the RAM cell number observed in these five accessions under stress treatment (Figure 4E,F). Furthermore, the RAM size changed as the result of different PD sizes in all the accessions and the TD size in L*er* (Appendix A). We also found that in all the accessions, the TD cell size was shorter under osmotic stress conditions than in control conditions, indicating that TD cells transit faster to the EZ (Appendix A). On the other hand, the length of the completely elongated cells was more affected in Col-0 and Cvi-0 as compared to the other accessions (Figure 4D). Interestingly, the Sg-2 accession was the least affected in both the RAM cell number, the elongation zone size, and the length of completely elongated cells (Figure 4D,E and Appendix A).

Surprisingly, and opposite to what has been reported before for Col-0, L*er*, and Ws [35,51,52], we did not find a direct correlation between the number of cells in the RAM and the length of the primary root of Cvi-0 and Ws accessions under our control conditions (Figure 4B,E). These data are consistent with a recent report in which Cvi-0 was analyzed [48].

Finally, and consistent with what we found in Col-0 (Appendix A), the SCN morphology of the four *Arabidopsis* accessions studied was not affected under hyperosmotic stress conditions (Appendix A).

### 3.5. The Stress Zone Is Not an Adaptive Response of Plants to Hyperosmotic Stress Conditions

When plants were subjected to a hyperosmotic acute condition, we observed an abnormal radial cell growth (that we have called the stress zone (SZ)) in the epidermis and the cortex, as has been reported previously [50] (Figure 2A). To address whether the SZ morphology could be an adaptive response to hyperosmotic conditions, we sowed all the *Arabidopsis* accession seeds directly under hyperosmotic conditions and found that, after 10 days of stress, the primary roots did not develop the SZ (Appendix A), despite cell morphology, root growth, and germination rate being affected (Appendix A). In addition, the SZ was also observed when we applied a hypoosmotic stress condition, transferring 6-dps seedlings that grew one day in the osmotic stress condition to our control medium and evaluated the response after two recovery-time periods: 8 and 12 h (Figure 5). We detected a SZ when we applied hyperosmotic or hypoosmotic stress conditions; in the first case, we found that the epidermal and cortex cells swelled as reported [31] and some accessions developed root hairs earlier (8 h; L*er* and Cvi-0) than the others (Figure 5). Under hypoosmotic conditions, the epidermal cells also swelled but to a lesser extent compared to the hyperosmotic condition; however, and opposite to what we expected, all the accessions developed root hairs after 8 h of transference to the hypoosmotic treatment. This might be counterintuitive because we originally thought that root hairs might be a response to the increase in solutes, to improve water and nutrient uptake, but it was rather a response to the sudden change in solutes, regardless of the direction of change, and more pronounced under hypoosmotic conditions (Figure 5). In addition, there was a wide natural variation to osmotic changes in the five accessions tested under both treatments; L*er* showed a prominent response to both treatments, whereas Col-0 and Cvi-0 had similar mild responses to both treatments, while Sg-2 barely responded to hyperosmotic stress conditions (both at 8 and 12 h), but had the same response as L*er* under hypoosmotic treatment. Finally, Ws was the accession with the smallest response to both treatments (Figure 5). These results demonstrate that the development of the SZ and the root hair development could be byproducts of the osmotic potential shock rather than a specialized root structure to face a hyperosmotic or hypoosmotic treatment or to changes in water availability.

### 3.6. Resilience to Osmotic Stress: Root Growth Arrest Is Transient and Reversible

To evaluate the resilience of cell behavior under osmotic stress, we tested the reversibility of cell growth and proliferation reduction under osmotic stress. We transferred seedlings subjected to stress treatment back to control conditions (see Material and Methods section for experimental procedures). After growing the seedlings for one day under hyperosmotic conditions with 300 mM mannitol, which produced a clearly detectable SZ, plants were transferred to our MS control conditions for three days. Although the stress impaired root growth after only one day of growth in hyperosmotic stress conditions (Figure 1), *Arabidopsis* seedlings were able to moderately recover normal growth rate and root meristem size (Appendix A). We also found that *CYCB1*; *1DB-GUS* expression levels partially returned to normal levels after transferring to control medium (Appendix A). This result shows that roots and cell behavior are resilient to transient stress conditions; first, being able to withstand and respond to high hyperosmotic stress conditions by adjusting cell proliferation, growth, and differentiation for a given time, and then being able to partially reestablish normal growth when transferred to non-stressful conditions.

### 3.7. Recovery after Strong Hyperosmotic Stress Conditions Does Not Correlate with the Osmotic Effect on Root Growth

The primary root growth of *Arabidopsis* has been useful to understanding proliferation and differentiation rate dynamics, either in control conditions or under different types of stress treatments. However, we also wanted to test whether the effect of hyperosmotic conditions on primary root growth and plant survival were related by growing plants in 400 mM of mannitol (−1.2 MPa), a strong hyperosmotic stress, for 18 days. Although all accessions showed some dry leaves and several lateral roots (Figure 6A), we were unable to predict at this moment which plants would resist and recover from the stress. Once the plants were transferred back to the control medium for 10 days for recovery, Sg-2 followed by Cvi-0 accessions recovered root length and shoots better than Ws and Col-0; whereas, L*er* plants did not recover from this stress and died (Figure 6B). The recovery of Sg-2 and Cvi-0 was reflected in their gain in fresh weight (Figure 6C). In addition, although the root growth of Sg-2 and Ws was less affected under hyperosmotic stress (Figure 4B), only Sg-2 showed a significant survival (Figure 6D). Meanwhile, Ws showed an intermediate surviving capacity (~15%), similar to Cvi-0. However, the root growth of Cvi-0 was more affected in hyperosmotic stress conditions than the one observed for Ws (Figure 4B). These results showed that Sg-2 and Cvi-0 were more tolerant to strong osmotic stress conditions, and that the recovery in control conditions or the survival capacity in soil after being subjected to a strong hyperosmotic stress conditions, do not always correlate with the initial effect of hyperosmotic stress conditions on primary root length.

### 3.8. Expression of Stress-Responsive Genes Does Not Correlate with Osmotic Stress Sensitivity

Osmotic stress responses involve changes in gene expression, including those that are related to water deficit, osmotic stress, or both conditions [53,54,55]. To evaluate whether hyperosmotic stress in root tissues induces the expression of genes involved in osmotic stress, we used two of the most tolerant accessions, Sg-2 and Cvi-0, and one of the most sensitive, Col-0, to measure the expression of osmotic stress-responsive genes. It has been shown that, under the osmotic stress treatment, the phytohormone abscisic acid (ABA) was induced and regulated the transcription of several genes that function in stress response and tolerance. We selected genes that are ABA-dependent such as *RAB18* (*RESPONSIVE TO ABA 18)*, *RD29B* (*RESPONSIVE TO DESICCATION 29B*), and *COR15A* (*COLD REGULATED 15A*); one gene that is ABA-independent: *RD29A* (*RESPONSIVE TO DESICCATION 29A*); and one gene whose coding product is involved in ABA biosynthesis: *NCED3* (*9-CIS-EPOXYCAROTENOID DIOXYGENASE 3*) [56,57,58]. Intriguingly, in control conditions the expression of *RAD29B* was higher in Cvi-0 and Sg-2 than in Col-0, and *NCED3* expression in Sg-2 was also slightly higher than Col-0. However, *COR15A* and *RAB18* showed similar levels in both tolerant and sensitive accessions, whereas the levels of *RD29A* were even lower in Cvi-0 and Sg-2 than in Col-0 (Figure 7A). These results suggest that in control conditions, the tolerant accessions do not have a generalized stress response. In other words, after 8 h of hyperosmotic stress, all of these genes strongly increased their expression in the roots of the three accessions. However, only *COR15A* in Sg-2 and *RAB18* in Cvi-0 showed slight but not significant increments in expression levels as compared to Col-0. The expression of the other genes in both Cvi-0 and Sg-2 was similar or even lower as compared to Col-0 (Figure 7B), indicating that the expression of these osmotic stress-responsive genes does not correlate with their phenotypic stress sensitivity. In addition, significant and opposite expression patterns between Sg-2 and Cvi-0 were observed for *COR15A*, *RAB18*, and *RD29A* (Figure 7B), but again without any correlation with their osmotic tolerance phenotype. Although we only quantified the expression of five stress-response genes and we still don’t know if their induction rate is key to the different sensitivities among accessions, these results suggest a rather complex adaptive evolution of the genetic network responsible for the natural variation of root responses to hyperosmotic stress, which deserves further attention in future studies.

## 4. Discussion

Water, which is taken up by the root system, is the most limiting resource for plant growth. Under water deficit, the root acts as a sensory system integrating changes in water content to respond accordingly. The osmotic adjustment occurs in the roots before the leaves to enhance turgor pressure for continued root growth and absorption of water and nutrients [59]. Therefore, we used the primary root of *Arabidopsis* to understand how it responds to hyperosmotic stress conditions and its variation in different accessions. This organ enabled us to perform in vivo quantitative cellular analyses of different *Arabidopsis* accessions to evaluate how cell proliferation and differentiation are affected individually under these conditions and how resilient they are once the stressful condition has been removed.

In our study, we exposed plants to an osmotic shock, changing drastically and immediately the osmotic pressure of the medium. In the 15 tested accessions root growth was impaired under our stress conditions, affecting both primary and lateral root growth, which is consistent with what has been previously reported for Col-0 [39,40,41]. Although different types of stress affect both primary and lateral root length, the latter is hypersensitive to salt stress in comparison to growth of the primary root [50]. According to Julkowska and collaborators, under salt stress conditions there are four root strategies to cope with salt stress using three different parameters: Primary Root (PR) growth, Lateral Root (LR) growth, and LR number. One of their strategies implied that the PR growth was more affected than LR growth under long-term exposure to ionic stress conditions [60]. Our geometric morphometric principal component analysis showed that the primary root length was the outlier to understand the impact of osmotic stress induced by mannitol in root architecture in the five accessions tested here. It could be very interesting to study the interplay between PR and LR growth in our experimental system during different developmental stages as it has been reported that the effect on the ratio of LR growth vs. PR growth changes with different developmental stages as well as with different experimental procedures [50,60].

In the primary root, the altered cellular patterns resulting from root growth under stressful conditions could be a useful experimental subject to approach to further unravel the role of different components on morphogenetic patterns in this organ. For example, under ionic stress conditions in *Arabidopsis* and under osmotic or water stress in maize, rice, and hybrid poplar, the cell expansion and cell production rate are affected, thus altering primary root growth [60,61,62,63,64,65]. The strength of the osmotic stress also affects each root domain differently, i.e., the meristematic cell number that is reduced in severe stress (−1.2 MPa), affecting the size of the *Arabidopsis* primary root [41]. Likewise, the cells at the TD are very sensitive to diverse environmental cues such as gravity, light, humidity, and various types of stress [24,66]. In maize and soybean, the relative elongation rate under water deficit is unaffected at the apical zone near the meristem, but it is inhibited throughout the elongation zone [22,67]. This occurs due to cell-specific structural changes and metabolic properties towards stress conditions in the different zones and domains along the longitudinal root axes [66,68].

Under our experimental conditions, the reduction in primary root growth could be explained by the decrease in the meristem cell number of the RAM and the shorter lengths of the differentiated cells in plants subjected to hyperosmotic stress conditions. Analogous to the results presented here, under ionic stress conditions, root proliferation, and elongation explained smaller RAM and EZ in the root [31,65,69]. In addition, RAM differentiates prematurely under water deficit in wheat root [70] and various species growing under diminished water availability; it has been suggested that this is an adaptive plant response to cope with this stress condition [70,71]. In our study we showed that the decrease in size of the RAM is related to a premature transit in two points: the proliferation domain to the transition domain, as the number of cells in the proliferation domain changes under osmotic stress treatments as compared to our control treatment; and the transition domain to the elongation zone, as the cell size in the former is shorter in almost all the accessions tested.

Additionally, we found that some root cells (cortex and epidermis) at the intersection between the RAM and the EZ showed a radially swollen phenotype when plants are exposed to a hyperosmotic shock condition. A similar phenotype was reported in wheat, maize, soybean, rice, and *Brachypodium* in response to osmotic and water stress conditions [31,70,72,73]. In wheat, swollen cells stop proliferating as they no longer stain with tetrazolium violet, and also increase their proline content, indicating an osmotic adjustment [70]. Furthermore, a drought-tolerant wheat cultivar has a lower percentage of swollen roots than another more sensitive cultivar [70]. In contrast to wheat cultivars, in *Arabidopsis* we found the same swollen cells in both sensitive and tolerant accessions. Moreover, our results indicate that this phenotype depends on the sudden osmotic potential change, which seems to be sparked by a lower (hypo) or higher (hyper) solute concentration shock on the medium, rather than an adaptive response to hyperosmotic conditions. Consistent with this observation, this swollen cell phenotype did not appear when plants were germinated and grown in 300 mM of mannitol, as these plants never experienced an osmotic shock.

Interestingly, this radial cellular expansion has been reported at the root apex of *Arabidopsis* under salt stress [72], in the lateral roots of plants under drought conditions [17], in plants with either altered cell wall biogenesis [74] or microtubule cytoskeletons [75], or multivesicular body biogenesis [76]. Given that swelling is determined by the physicochemical properties of the cell wall [77], its occurrence is not surprising in the transition domain cells that are located between the RAM and the EZ in wheat [70]. In *Arabidopsis*, the transition domain has unique physiological properties such as alterations in their cell wall structure and vacuolization that enables fast length growth in the EZ [24,25,66]. In contrast, cell morphology of the root SCN was unaffected by osmotic stress and it has been reported that the maintenance of functional SCN is an ability of the roots to withstand the concurrent environmental conditions, which allows roots to restore their growth dynamics when conditions are more favorable [78,79].

According to our results, root cell proliferation is resilient, and once the hyperosmotic stress is withdrawn, normal growth partially recovers after one day, even if plants have been growing under highly stressful conditions such as 300 mM of mannitol. Although the root growth of all accessions tested here was affected, each accession had a different sensitivity, Sg-2, Ws, and Cvi-0 being less affected than Col-0 and L*er*. The recovery of root growth after strong osmotic stress (400 mM for 18 days) was also variable, since Sg-2 and Cvi-0 exhibited more tolerance than the rest of the accessions. Puzzlingly, these responses contrast to what has been previously reported for salt stress in aerial tissues, where Cvi-0, Sg-2, Col-0, and L*er* were the most sensitive accessions [12,80]. Although we still do not know if these differences are due to the type of stress, we cannot rule out the possibility that the root and aerial tissues respond differently to osmotic stress. However, our results suggest that integrative studies that consider both the shoot and the root are crucial to define stress sensitivity. On the other hand, the resilience of Sg-2 was surprising, since its location of origin (Sankt Georgen, Germany) is a cool and wet climate, and yet it has been reported as a salt stress-sensitive accession [12] raising questions about the adaptive nature of its phenotype. In contrast, Cvi-0 is native to a warm and dry climate [81,82] and more resistant to various types of stress [82,83,84], which might be expected since Cvi-0 is believed to be the result of a relatively recent introduction into the African continent, reflecting evolutionary bottlenecks, drift, and adaptive evolution.

The specific mechanism that Cvi-0 and Sg-2 use to resist the strong osmotic stress is still unknown. Although both accessions are able to induce some osmotic stress genes, the induction levels were similar or even lower than the ones we obtained with Col-0, a sensitive accession; therefore, other mechanisms could be involved. It has been reported that Cvi-0 has higher ABA levels than Col-0 [84,85], which is related to the higher levels of *RD29B*, a gene mainly controlled by ABA, observed in Cvi-0 as compared to Col-0, under control conditions. *RD29A* decreased its expression in Cvi-0, but it is mainly induced through the ABA-independent pathway [86,87,88,89]. The increase of *RD29B* and *NCED3* observed in Sg-2 could also be related to high levels of ABA. Therefore, ABA content or a constitutive activation of stress responses may be responsible for generating different sensitivities to osmotic stress, rather than changes in expression in some genes.

This work has uncovered quantitative cellular data that helps to explain how root development is affected by osmotic stress conditions, the natural variation on the plasticity of these mechanisms, and how these organ responses relate to tissue growth dynamics, cell proliferation, and differentiation. It has also been shown that this experimental system of altered growth and cellular dynamics may be a useful system to test the models of coupled cell proliferation and physicochemical properties in order to understand the emergence of cellular patterns and behaviors in complex organs, such as the *Arabidopsis* root [90]. Future experimental and “*in silico*” approaches will be necessary to further unravel the developmental and genetic nature of stress responses and their evolution.

## 5. Conclusions

In this study we showed that, in all the *Arabidopsis* accessions used, the primary root growth diminished under stress conditions. This root reduction is dependent on the RAM cell number and the length of the completely elongated cells. Interestingly, we showed that the morphology of the stem cell niche does not change under hyperosmotic stress conditions in *Arabidopsis* roots. In addition, we found that the cell swelling that appears under these stress conditions is not a root response to hyperosmotic stress, but rather a reaction to an osmotic shock treatment, with either higher or lower solutes in the growth medium. Moreover, PCA analysis showed that the primary root length explains almost 90% of the variation found among accessions. Finally, the sensitivity to osmotic stress was different in each accession, Sg-2 and Cvi-0 being more tolerant than Col-0 and L*er*. In conclusion, these results pave the road to the cellular and whole-organ characterization of osmotic stress responses in the roots of *Arabidopsis* accessions, opening an avenue for larger-scale phenotype characterizations towards the mapping of the underlying genetic bases of osmotic stress responses.

## Figures and Tables

**Figure 1 genes-10-00983-f001:**
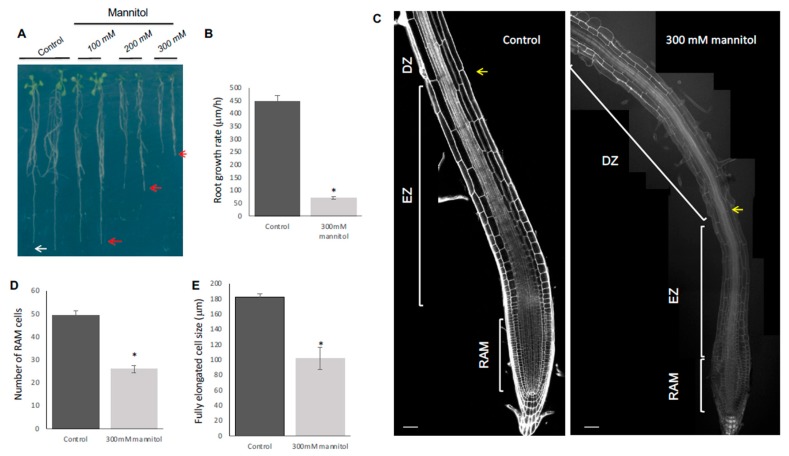
Osmotic stress conditions yield shorter roots of Col-0 plants. (**A**) Representative images of the root growth under osmotic stress for 8 days. (**B**) Root growth rate under control and hyperosmotic conditions for 1 day at 300 mM mannitol. (**C**) Median longitudinal confocal images of roots grown during 5 days in MS and then transferred to control conditions or treated with hyperosmotic conditions for 1 day (300 mM mannitol); the different apical–basal zones are shown. RAM for root apical meristem; EZ for elongation zone, and DZ for differentiation zone. The yellow arrow marks the first epidermal cell with a hair root. Roots were stained with propidium iodide. The white bar represents 50 µM. RAM cell number (**D**) and size of fully elongated cells (**E**) of roots under control and hyperosmotic conditions for 1 day at 300 mM mannitol. Values are mean ± SEM (n ≥ 20) * *p* < 0.05, according to Student’s *t*-test.

**Figure 2 genes-10-00983-f002:**
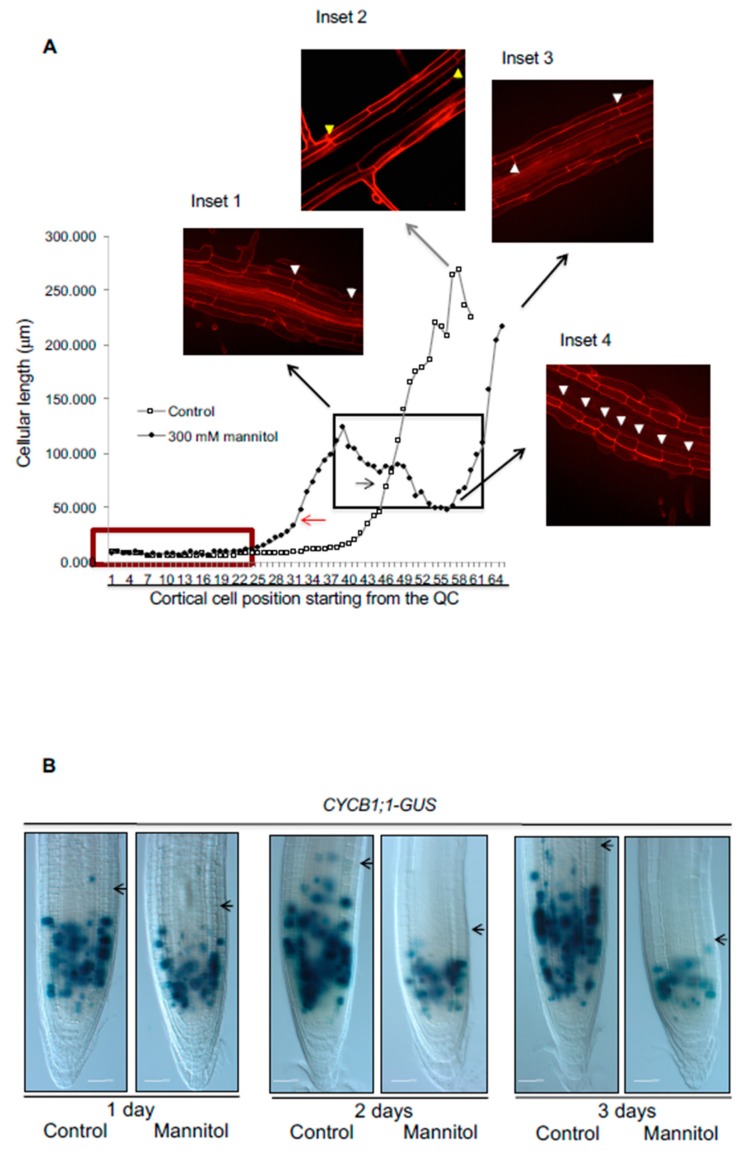
Hyperosmotic stress reduces growth of primary root by affecting cell proliferation and cell elongation, and also generates a radial enlargement of epidermal cells in the Stress Zone (SZ). (**A**) Growth curve of average cell length of root cortical cells considered from the quiescent center (QC) to the maturation zone of untreated Col-0 seedlings (control) or Col-0 seedlings treated for one day with 300 mM mannitol (mannitol). The insets are confocal images of control roots (inset 2) or roots with hyperosmotic stress (insets 1, 3, and 4). Inset 1 shows fully elongated cells localized before the stress zone in the stress conditions; inset 2 marks fully elongated control cells; inset 3 shows fully elongated cells after the stress zone in stress conditions and inset 4 marks cell length in the stress zone in stress conditions. We use yellow (control) or white arrowheads (mannitol) to indicate the size of the cells at different points of the cell length curves. In the plot, meristematic cells are indicated with a red box and arrows indicate the transition of cells from the RAM to the EZ in mannitol (red arrow) and control conditions (black arrow). (**B**) Five-day-old *CYCB1*; *1DB-GUS* seedlings germinated on MS medium were transferred to control conditions or media supplemented with 300 mM mannitol for 1 to 3 days (n ≥ 20). Black arrows indicate the end of the meristem in each case. Bar = 50 μm.

**Figure 3 genes-10-00983-f003:**
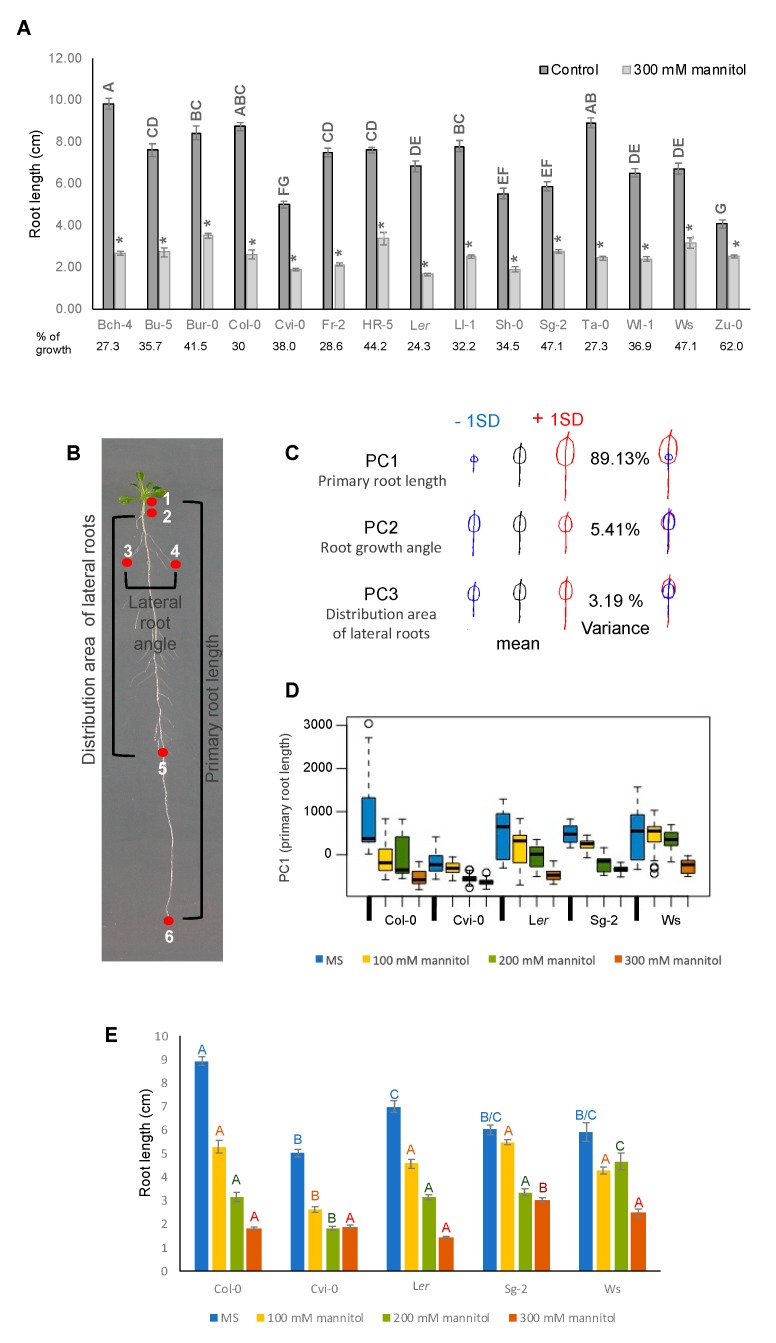
Hyperosmotic stress affects primary root growth of different *Arabidopsis* accessions. (**A**) Effect of hyperosmotic stress conditions on the root length of Bch-4, Bu-5, Bur-0, Col-0, Cvi-0, Fr-2, HR5, L*er*, Ll-1, Sh-0, Sg-2, Ta-0, Wl-1, Ws, and Zu-0 *Arabidopsis* accessions. Data are means ± SEM (n = 25–30) of three biological replicates. Different letters indicate significant differences between root lengths of accessions under control conditions and (*) points out the length difference under the 300 mM mannitol condition of each accession in relation to its own control (*p* < 0.05, two-way ANOVA followed by Tukey’s multiple comparisons test). (**B**) Landmarks used for principal component analysis of seedlings treated with osmotic stress conditions. (**C**) Principal component (PC) 1 (primary root growth) captured more than 80% of the variation of six accessions (Col-0, Cvi-0, L*er*, Sg-2, Ws, and Zu-0). For the morphometric analysis we used the Shape Model Toolbox software, implemented for roots as RootScape. The model is made out of 20 landmarks, as follows: one landmark at the base of the root (1), one at the tip of the root (6), two at the locations of the first and last elongated lateral roots along the primary root respectively (2 and 5), and two at the widest points of elongated lateral roots at each side of the primary root (3 and 4); 14 pseudo-landmarks placed evenly spaced between the landmarks build a polygon that captures the convex hull-shape of the root architecture. Here we only showed the three PCs that capture the 97.7% of the variation as an arbitrary cutoff. (**D**) Boxplot of the PC1 (primary root length) allometric axis, showing the accessions and treatments; a higher PC1 value corresponds to longer roots. In all accessions, PC1 values were higher in the control treatment as compared to the osmotic stress treatments. (**E**) Root lengths of Col-0, Cvi-0, L*er*, Sg-2, and Ws in control conditions and 100, 200, and 300 mM mannitol. Values are mean ± SEM (n = 25–30), the statistical analyses were performed between each treatment (MS, 100 mM mannitol, 200 mM mannitol, or 300 mM mannitol), *p* < 0.0001, according to the two-way ANOVA test.

**Figure 4 genes-10-00983-f004:**
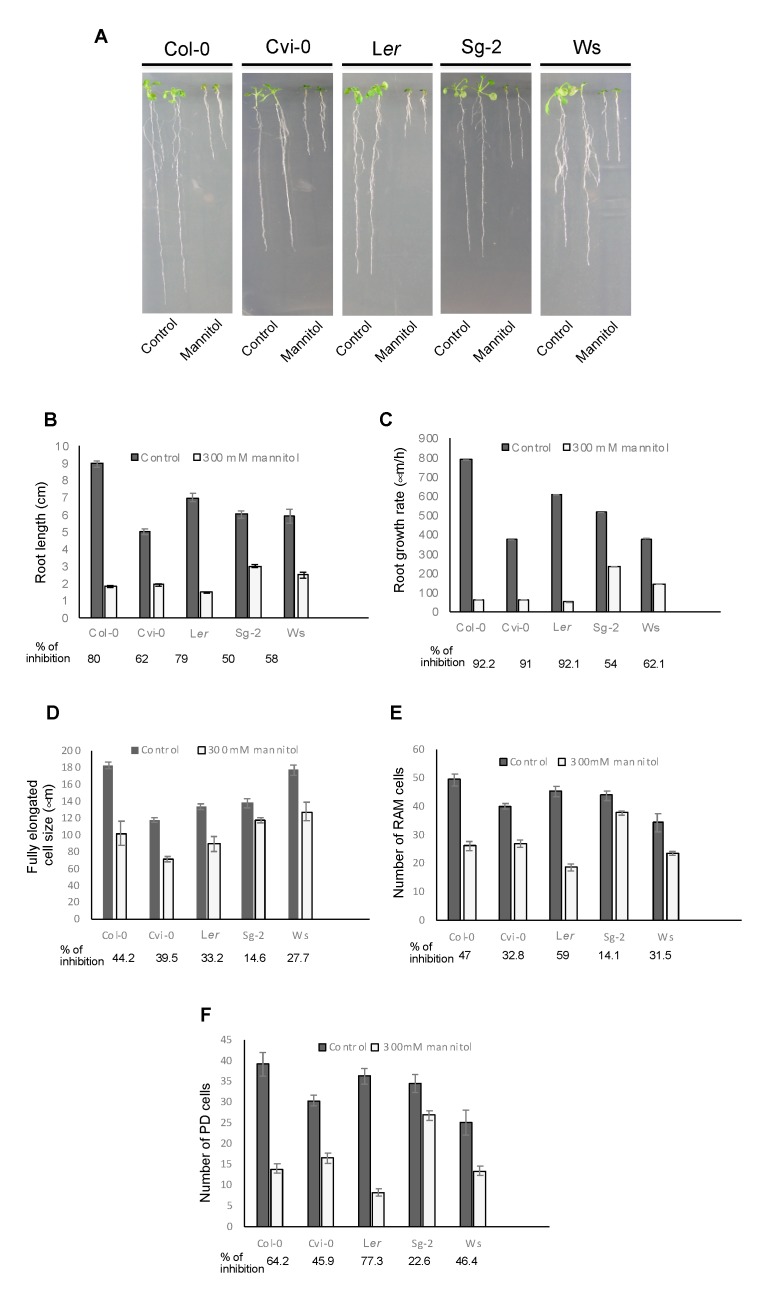
Diversity in primary root length and cellular parameters of five different *Arabidopsis* accessions growing in control and osmotic stress conditions. (**A**) Representative images of Col-0, Cvi-0, L*er*, Sg-2, and Ws seedlings, grown for 5 days in control conditions and then transferred to control or mannitol 300 mM for 8 days. (**B**–**F**) Different quantitative cellular parameters of the accessions treated as in **A**. The growth rate in **C** was measured from day 5 to day 6 after stress treatment. Values are mean ± SEM (n = 25–30).

**Figure 5 genes-10-00983-f005:**
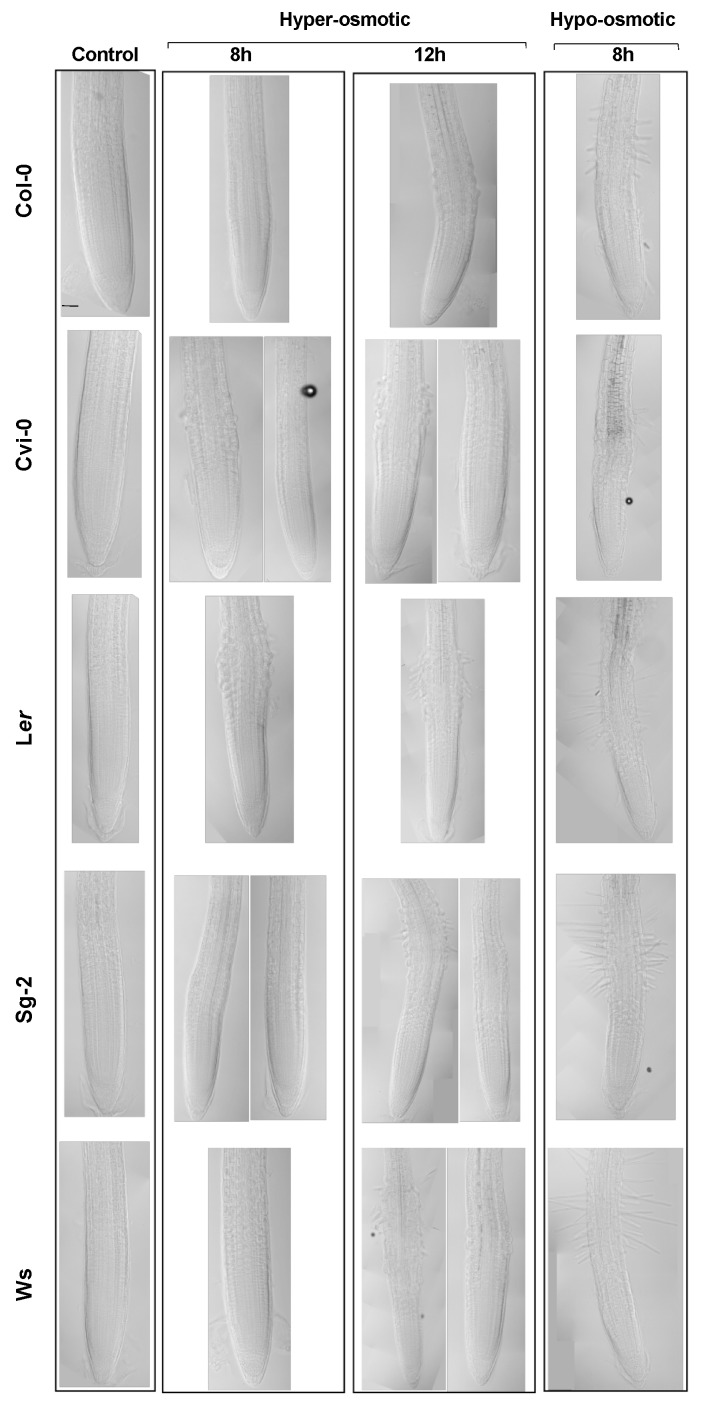
The Stress Zone (SZ) is formed in both hyperosmotic and hypoosmotic stress conditions. Median longitudinal images of the primary root of plants grown for 5 days in the control condition and then transferred to the control medium (Control), or transferred to 300 mM mannitol for 8 or 12 h (hyperosmotic stress), or transferred to 300 mM mannitol for 24 h and then transferred to the control medium for 8 h (hypoosmotic stress). Note that in some treatments, the accessions had two different stress zone morphologies, which are represented here. Bar scale = 50 µm.

**Figure 6 genes-10-00983-f006:**
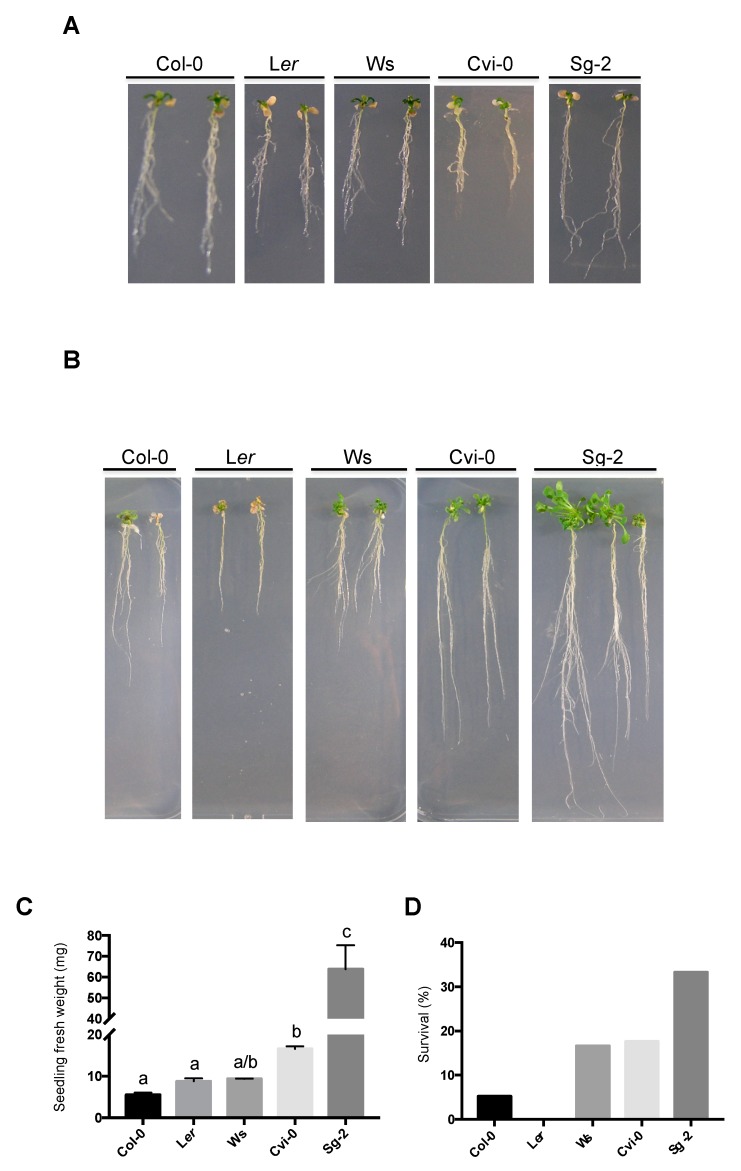
Plant recovery of the five *Arabidopsis* accessions after hyperosmotic stress condition. (**A**) Representative images of 5-dps plants transferred to 400 mM mannitol for 18 days (note that each week, plants were relocated to a fresh medium with 400 mM mannitol). (**B**) Seedling from (**A**) grown for 10 days in control conditions after being 18 days in 400 mM mannitol. (**C**) Fresh weight tissue (mg) from the seedlings shown in (**B**). ANOVA (Kruskal-Wallis) test *p* < 0.01, (n > 15). (**D**) Survival analyses of seedlings from (**A**) grown for 15 days in soil (n > 15).

**Figure 7 genes-10-00983-f007:**
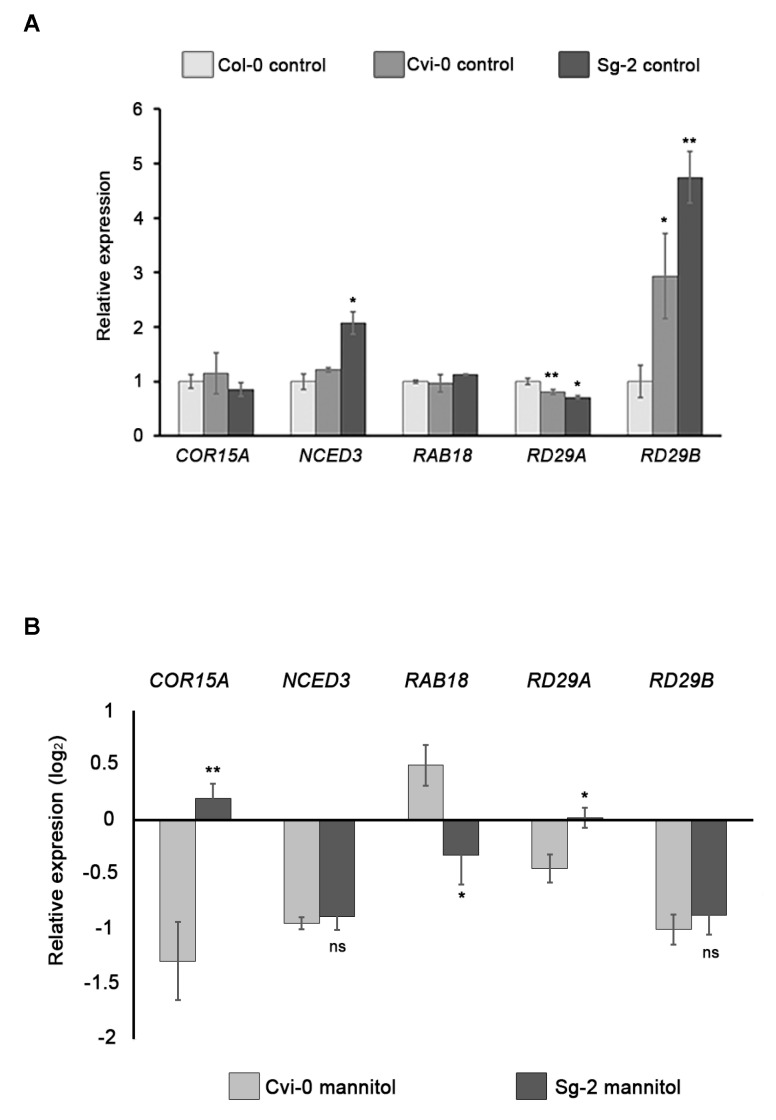
Expression analysis of stress-responsive genes in Col-0, Cvi-0, and Sg-2 accessions. (**A**) Gene expression of *COR15A*, *NCED3*, *RAB18*, *RD29A*, and *RD29B* in roots under control conditions. Asterisks indicate statistical significance with respect to Col-0, determined by Student’s *t*-test, * *p* < 0.05; ** *p* < 0.005. (**B**) Relative expression of the abovementioned genes, showing log2 fold changes relative to Col-0 mannitol (hyperosmotic stress). Asterisks indicate statistical significance with respect to Cvi-0 mannitol, determined by Student’s *t*-test (ns, no significant; * *p* < 0.05; ** *p* < 0.005). Bars show means ± SEM of three biological replicates.

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
