# Peer review of "Natural Root Cellular Variation in Responses to Osmotic Stress in Arabidopsis thaliana Accessions"

_genes, 2019, doi:10.3390/genes10120983_

Round 1
Reviewer 1 Report
I acknowledge the work made by the authors that significantly increase the quality of the ms. Nevertheless, I would like to bring again to their attention the conclusion that draw for fig 7. Authors state that there is no correlation between tolerance phenotype and the set of gene tested. This true, but only for the time point tested (8hours after stress application). Such strong conclusion needs to be supported by kinetic analysis of gene expression. Indeed, the differential in expression between control and treatment would likely vary as function of time, and maybe tolerant plant are able to induce faster their gene expression that the other accesion or vice versa. I heard that the author does not want to do the experiment, but I think that a sentence showing that they are aware about this weakness should be mention in the text.
I have some minor point regarding figure legend 3. What the letters A, B, C … means. If it is per wise mean comparison between which treatment is it made. From what I can understand it seems to between genotype within a single treatment. Maybe the graph could be drawing in another way. For the fig5, what is written in the figure and in the figure legend is not fitting each other.
Author Response
Decision: Pending minor revisions
Marko Kozlina said: Please cite Figure S4 in the main text during your revisions. Thank you for this observation because some Figures were not properly cited. Figure S4 was already cited, please see line 252 and 257.
Referee: 1
I acknowledge the work made by the authors that significantly increase the quality of the ms. Nevertheless, I would like to bring again to their attention the conclusion that draw for fig 7. Authors state that there is no correlation between tolerance phenotype and the set of gene tested. This true, but only for the time point tested (8 hours after stress application). Such strong conclusion needs to be supported by kinetic analysis of gene expression. Indeed, the differential in expression between control and treatment would likely vary as function of time, and maybe tolerant plant are able to induce faster their gene expression that the other accessions or vice versa. I heard that the author does not want to do the experiment, but I think that a sentence showing that they are aware about this weakness should be mention in the text.
Thank you for this observation; we have added a sentence to point out the importance of induction rate in the stress responses of the different accessions, see line 496-498
I have some minor point regarding figure legend 3. What the letters A, B, C … means. If it is per wise mean comparison between which treatment is it made. From what I can understand it seems to between genotype within a single treatment. Maybe the graph could be drawing in another way.
Thank you. The text in the figure legend has been modified to explain the meaning of the letters in both 3A and 3E and we added a new plot (3D) in this Figure.
For the fig5, what is written in the figure and in the figure legend is not fitting each other.
Thank you for this comment, the mistake was corrected and for the best understanding of this figure, we first show the hyper-osmotic stress photographs and then hypo-osmotic pictures. In addition, in this new figure we also corrected another mistake that we had for the Ler accession at 12hr under hyper-osmotic conditions. In the previously Figure, we showed a photograph of the Ler accession with a hypo-osmotic treatment instead of a photograph with a hyper osmotic condition; in the new figure we have corrected this.

Reviewer 2 Report
The research by Cajero-Sanchez et al is adequately planned and presented. The methods are well and properly designed and the results are clearly presented. There are only two points I would like to highlight:
the authors should explain why they selected the specific genes for QPCR analysis and I have some concerns related to the morphological analysis and the statistical data following morphological analysis. Ideally the authors should use stereological approaches which include statistical analysis and provide an accurate morphological assessment.
Author Response
Referee: 2
The research by Cajero-Sanchez et al is adequately planned and presented. The methods are well and properly designed and the results are clearly presented. There are only two points I would like to highlight:
the authors should explain why they selected the specific genes for QPCR analysis
In the new version, we have added a paragraph justifying why we selected the five genes, please see line 475-484.
and I have some concerns related to the morphological analysis and the statistical data following morphological analysis. Ideally the authors should use stereological approaches which include statistical analysis and provide an accurate morphological assessment.
Thank you for this observation, we have included a new plot on Figure 3 (3D) with the allometric axis analysis.
